# Bio-Based Aerogels in Energy Storage Systems

**DOI:** 10.3390/gels10070438

**Published:** 2024-06-30

**Authors:** Vilko Mandić, Arijeta Bafti, Ivana Panžić, Floren Radovanović-Perić

**Affiliations:** Faculty of Chemical Engineering and Technology, University of Zagreb, Trg Marka Marulića 19, 10000 Zagreb, Croatia; ipanzic@fkit.unizg.hr (I.P.); fradovano@fkit.unizg.hr (F.R.-P.)

**Keywords:** aerogels, energy storage systems, battery electrode materials, battery separators, supercritical fluid drying, freeze-drying, cellulose, bio-based materials

## Abstract

Bio-aerogels have emerged as promising materials for energy storage, providing a sustainable alternative to conventional aerogels. This review addresses their syntheses, properties, and characterization challenges for use in energy storage devices such as rechargeable batteries, supercapacitors, and fuel cells. Derived from renewable sources (such as cellulose, lignin, and chitosan), bio-based aerogels exhibit mesoporosity, high specific surface area, biocompatibility, and biodegradability, making them advantageous for environmental sustainability. Bio-based aerogels serve as electrodes and separators in energy storage systems, offering desirable properties such as high specific surface area, porosity, and good electrical conductivity, enhancing the energy density, power density, and cycle life of devices. Recent advancements highlight their potential as anode materials for lithium-ion batteries, replacing non-renewable carbon materials. Studies have shown excellent cycling stability and rate performance for bio-aerogels in supercapacitors and fuel cells. The yield properties of these materials, primarily porosity and transport phenomena, demand advanced characterization methods, and their synthesis and processing methods significantly influence their production, e.g., sol–gel and advanced drying. Bio-aerogels represent a sustainable solution for advancing energy storage technologies, despite challenges such as scalability, standardization, and cost-effectiveness. Future research aims to improve synthesis methods and explore novel applications. Bio-aerogels, in general, provide a healthier path to technological progress.

## 1. Introduction

Even though this article is based on bio-based aerogels, it should first be clarified that aerogels can be categorized into two main types, inorganic and organic, with further subdivision based on the materials used in their gel structure design [1]. Interestingly, recent attention seems to be going in the direction of biodegradable and bio-based polymers due to their potential to reduce environmental impact [2]. Namely, these porous materials offer versatile properties and can be tailored through specific synthesis routes for a broad range of applications. For instance, polysaccharide-based aerogels have been utilized for environmental engineering, construction, medical practice, packaging, electrochemistry, and other [3]. However, bio-based aerogels are primarily employed as adsorbent materials and drug-delivery systems [4]. Their advantages include many parameters common with other aerogels, such as mesoporosity and high specific surface area, as well as their biocompatibility and biodegradability, which are particularly advantageous in the nowadays-relevant context of environmental friendliness. Among those, cellulose, lignin, and chitosan, are the most utilized and studied materials due to their availability, recyclability, and excellent properties [5].

Traditional energy storage, like battery devices, typically consist of three essential components: electrodes, electrolyte, and separator. The fundamental working principle involves ions migrating between electrodes, facilitated by the separator, during the charge/discharge processes, thereby generating energy and power. Electrochemical energy is primarily stored at the interfaces of the electrodes and the electrolyte through faradaic and non-faradaic reactions [6]. Rechargeable batteries (mostly referring to lithium-ion batteries (LIBs)), supercapacitors (SCs), and fuel cells (FCs) have received significant attention due to their high energy densities and extended lifespans [7,8,9]. This review paper will focus on these devices.

Energy and power density, rate capability, and the cycle life of batteries, fuel cells, and supercapacitors vary widely depending on the materials used for electrode and electrolyte [10]. Therefore, the selection of renewable sources is crucial for designing high-performance energy materials to expand their practical applications. While solid-state batteries have seen significant development, lithium-ion batteries remain the most widely used due to their favourable characteristics, such as ionic conductivity and thermal and electrochemical stability. Electrochemical energy storage systems commonly rely on carbon materials known for their well-designed porous microstructure, high electrical conductivity, and good mechanical properties [11,12]. In LIBs, graphite is the prevalent choice for the anode, while activated carbons with large specific surface areas or porous carbons serve as electrodes in electrolytic capacitors and conductive substrates in lithium–sulphur (LIS) batteries [13,14]. However, the synthesis of high-quality carbon compounds typically involves complex and costly procedures, high temperatures, non-renewable precursors, and firm chemical conditions, limiting their commercial viability [15,16,17]. Thus, there is a critical need to develop synthetic routes for the facile, efficient, and environmentally friendly preparation of carbon material using low-cost, sustainable precursors. Lignocellulosic biomass emerged as a promising candidate for this purpose, given its abundance, adaptability, sustainability, and cost-effectiveness. Researchers have therefore focused on utilizing cellulosic materials to fabricate electrodes with high energy and power density, lightweight current collectors, and functional separators leveraging the advantageous properties of cellulose for electrochemical energy storage and conversion devices [18,19,20,21]. LIBs offer high energy density, whereas SCs excel in high power density, cyclability, and stability. The evolution of energy storage devices hinges on advancements in their key components, namely liquid electrolytes, separators, and electrodes. Liquid electrolytes’ progress centres on enhancing ionic conductivity and thermal/electrochemical stability [22], while separator development is linked to reducing ionic resistivity and enhancing overall safety [23,24]. The stability and electrochemical performance are related to the improvement of the electrodes. Carbon-based electrodes have garnered substantial attention due to their porous morphologies, facile modification, high specific capacitance, and electrochemical stability [25,26]. These electrodes are derived from a variety of petroleum-based chemicals such as poly(vinyl alcohol) (PVA) [27], polyethene oxide (PEO) [28], polyvinylidene fluoride (PVDF) [29], polyacrylonitrile (PAN) [30], PEDOT-PSS [31], etc. PAN was recognized as the most prevalent precursor owing to its high carbon yield, quality carbon structure, and formation of diverse nanostructures and morphologies. Researchers have synthesized novel PAN-based nanostructures, including flower particles, hollow nanoparticles, ultrafine nanofibers, and 2D nanostructures, each offering unique surface and physical properties ideal for energy storage applications [32,33,34,35,36]. For instance, PAN-based cloth electrodes in SCs demonstrated a high energy density of 4.03 Wh kg^−1^, with excellent cycling stability over one thousand cycles [37], while random PAN-based carbon nanostructures in lithium–sulphur batteries exhibited an initial reversible capacity of 840 mA kg^−1^, along with high cycling stability over 150 cycles [38]. A huge focus is devoted to different PEDOT:PSS materials embedded with 2D nanomaterials for flexible and wearable energy storage materials. Different competing materials such as polyaniline, PEDOT-PSS, PAN, and most other carbon precursors are derived from non-renewable and environmentally toxic petroleum sources, necessitating the exploration of alternative, sustainable materials. [31] A promising alternative lies in renewable, bio-based carbon precursors, which offer environmental sustainability, cost-effectiveness, and potentially enhanced electrochemical performance. Leveraging biomass and biowaste materials for carbon electrodes aligns with the broader concept of biorefinery, which aims to convert bio-based resources into value-added products [39]. Thus, the pursuit of high-performance nanostructured bio-based carbon aerogels for energy storage applications represents a promising way for sustainable technological advancement [40].

The preparation methods and diverse applications of aerogels have undergone extensive investigation, as evidenced by numerous review articles. However, there exists a notable scarcity of literature focusing specifically on the utilization of bio-based aerogels in energy storage devices. This review aims to bridge this gap by exploring recent research on the synthesis of bio-based aerogels and their potential applications in energy storage systems. Therefore, this review will further elaborate on recent advantages, challenges, and future perspectives for the case of several divisions: according to the main types of energy storage devices, i.e., main battery types and supercapacitors,according to the preparation and characterization ability for the case of different types of bio-based aerogels,according to the suitability of the derived bio-based aerogels for different segments of the energy storage devices.

## 2. Bio-Based Aerogel Preparation, Characterization, and Types

### 2.1. Bio-Based Aerogels Preparation

Since the inception of aerogels, significant attention has been directed toward organic and metal oxide variants. However, inherent drawbacks such as hazardous degradation byproducts, poor biodegradability, and toxic precursors have spurred a shift towards the investigation of bio-based aerogels. These bio-based aerogels are primarily derived from proteins, polymers, and polysaccharides sourced from biomass [41]. While bio-based aerogels may not fully replicate all the advantageous properties of their inorganic counterparts, recent research has focused on developing hybrid aerogels tailored to specific applications. Among the various methods employed for the preparation of bio-based aerogels, the sol–gel method stands out as conventional. This method enables the formation of a highly porous 3D network and facilitates control of porosity, mechanical strength, specific surface area, and other final properties of the aerogels. Additionally, the inclusion of inorganics during preparation can impact the chemistry, but more importantly, can impact the pore size and pore size distribution within the aerogels. However, among all the parameters affecting aerogel properties, the drying process exerts the most significant influence—schematically shown in Figure 1(a). 

There are three main options: Ambiental drying—will enable removing the solvent residuals but will not enable the preparation of aerogel bodies due to the inability to mitigate capillary forces. Ambiental drying is facile but unsuitable for advanced applications.Freeze drying—will be more efficient for removing the solvents but will face optimization limitations for achieving various morphologies and processing samples of various sizes. Freeze drying, i.e., lyophilization is a more powerful, but not particularly versatile, method (schematically shown in Figure 1(b)).Supercritical drying—will enable efficient solvent removal and a variety of optimization possibilities for achieving various morphologies. Supercritical fluid drying technologies are the ultimate aerogel processing tools (schematically shown in Figure 1(b)).When talking about preparation of the active materials for energy storage devices, i.e., electrodes, pyrolysis of the bio-based aerogels is the crucial step for making them adequate. High-temperature pyrolysis treatment under the protection of inert gases, such as nitrogen and argon, is crucial to make them pure carbonous materials, without oxide implementation. A few pyrolysis regions can be separated:Around 200 °C, where chemically bound water is removed from the material.Below 300 °C, where the cleavage of intra- and inter-molecular hydrogen bonds occurs, while intra-molecular dehydration is predominant.Above 300 °C, where inter-molecular dehydration takes place along with decarbonylation, ring-opening, and aromatization.At 430 °C, while dehydration is nearly complete, and small aromatic clusters undergo significant deoxygenation and condensation between 430 and 650 °C.Beyond 650 °C, where the dominant reaction shifts from deoxygenation to dehydrogenation, resulting in highly aromatic chair with large aromatic systems composed of over six fused ring structures.Typically, the pyrolysis temperature for bio-based aerogels exceeds 800 °C to achieve the desired properties after sintering occurs, influencing electrical properties and the specific surface area of the samples, depending on the used precursor and the pyrolysis conditions.

### 2.2. Bio-Based Aerogels Characterisation

Considering the inherent porosity of the aerogels and their intended application in energy storage systems, it is noteworthy to talk about suitable characterization techniques. Namely, the chemisorption and physisorption pool of methods have been thoroughly employed to describe the morphology and texture of the aerogels, including bio-based aerogels. However, information regarding the systematization of the electrical properties according to the pore size and pore size distribution is not widespread. Therefore, bio-based aerogels display several typical pore sizes and pore size distribution behaviours, but in the matter of the topic of this review paper, for the potential efficient application in energy storage devices, pores should mostly be in the microporous range, below 2 nm, followed with mesopores, and thus have a multimodal pore size distribution.

Their electrical properties can be described using methods such as the following:Cyclic voltammetry (CV) and linear sweep voltammetry (LSV), which are usually used for describing charge and discharge capacities, as well as for the determination of Coulombic efficiency by measuring the potential–intensity curves in a potential range. Different oxidation and reduction peaks could be observed. Rate capabilities at different current densities and charge/discharge profiles could be presented, from which one can obtain the endurance test, as well as the average Coulombic efficiency after a certain number of cycles.Solid-state impedance spectroscopy (SS-IS) and electrochemical impedance spectroscopy (EIS) are used for further investigation of the interlayer effect on single-cell performance. The measurement system for measuring solid-state impedance spectroscopy consists of an impedance analyser, a BDS cell in which the sample is placed between two electrodes, a cryostat system for temperature control, and a computer (equipped with appropriate software). The real and imaginary components of the complex impedance are measured using an impedance analyser over a wide frequency and temperature range. The experimental spectra of the complex impedance are analysed equivalently to results given by EIS, by modelling an equivalent circuit using the complex nonlinear least squares (CNLS) method. Generally, SS-IS is used for determination of the electrical characteristics of individual constituents of energy-storage system. Usually, results are presented as Nyquist plots where different contributions and mechanisms of electric conductivity can be seen. Different charge transfer processes can be visible at the mid-frequency range of the plot, representing the interface process between the electrolyte and electrode. Resistance values at the end of the measurement are in agreement with the corresponding cycling performances of the cell.

Accordingly, resistance values of lithium-ion batteries depend on many factors, such as humidity, temperature, and state of health of the battery, so there is no typical range for all conditions. For example, at 30 °C battery cells with a health level greater than 80% should be in the range of approximately 50 mΩ [42]. On the other hand, charge capacities for different bio-based materials as electrode materials are in the range of 20 to 80 mAh g^−1^ [43].

The mechanical properties of aerogels are very important and require a separate review paper. However, at least for the case of the bio-based aerogels the threshold values for mechanical properties do not differ considerably. For energy storage devices, electrical behaviour may be more relevant.

### 2.3. Polysaccharide-Based Aerogels

Given the focus of this review on polysaccharide-based aerogels, particular attention will be given to the polysaccharide materials commonly employed in energy storage applications. SEM images of the chosen bio-aerogels are presented in Figure 2.

#### 2.3.1. Lignin

Lignin is a natural polymer composed of three phenylpropane units with an empirical formula C_31_H_34_O_11_, mainly consisting of coniferyl alcohol, sinapyl alcohol, and a low amount of p-coumaryl alcohol [44]. The structure and concentration of lignin vary significantly depending on factors such as botanical source, plant tissue, age, and type of cell wall layers. Consequently, different sources and isolation methods yield lignin with varying ratios and content of structural units, leading to distinct properties. As an inexpensive byproduct of the pulp and paper industry, lignin extraction from approximately 1013 tons of lignocellulosic biomass yields approximately 100 billion tons of lignin through various methods [45]. Despite its variability, lignin serves as a promising alternative to conventional polymers, offering a sustainable solution to reduce reliance on non-renewable fossil resources [46]. Due to its rich aromatic content, lignin presents an attractive raw material for the production of high-quality carbon with considerable yields [47,48]. Recent studies have demonstrated successful utilization in the production of structural carbon fibres and electrospun carbon mats [49,50,51], holding promising prospects for structural and electrical applications. Moreover, lignin-derived activated carbon exhibits comparable specific surface area and pore volume to commercial counterparts [52].

#### 2.3.2. Cellulose

Cellulose, another essential plant component, can be prepared as nanofibers through the fibrillation of 2,2,6,6-tetramethylpiperidine-1-oxyl radical (TEMPO)-mediated oxidation, resulting in TEMPO-oxidized cellulose nanofibers (TOCNFs) with a small width (mostly less than 3 nm) [53]. Furthermore, their slender fibre width facilitates the generation of additional meso- and micropores during carbonization, culminating in carbon aerogels with a large specific surface area advantageous for gas capture and energy storage applications. Cellulose aerogels offer several advantages, including regeneration, low-carbon footprint, low density, high porosity, and robust adsorption capabilities. They serve as precursors for porous carbon materials, making them particularly promising for electrodes with significant development potential [54,55]. Current research focuses on carbon materials with superior electrical conductivities and high specific surfaces, including carbon nanotubes, graphene, and carbon black [56,57,58]. However, their complex preparation, high costs, and non-renewable nature hinder industrial production. The 3D network structure resulted in excellent porosity, enhanced by secondary pores formed during carbonization preserving the network structure. The resulting porous material exhibits good electrical conductivity and large specific surface area. The 3D pore structure facilitates the insertion of lithium ions, increasing the number of active sites for ion transmission. Employing cellulose aerogels for preparing carbon-based materials with excellent cycling and rate performance offers a novel approach to developing low-cost, environmentally friendly lithium batteries [59]. 

#### 2.3.3. Chitosan

On the other hand, chitosan (CH), a renewable nitrogen-containing natural biopolymer, serves as an advantageous precursor for incorporating nitrogen functionalities into graphene-based electrodes [60]. It facilitates the entrapment of metal oxide nanoparticles within the graphene framework and acts as a gelation-cum-reducing agent for graphene oxide (GO) precursors However, it is worth noting that most electrode studies are conducted using a 3-electrode configuration rather than a 2-electrode setup, which limits their practical applicability [61,62]. Moreover, the supercapacitor performance is also directed by the nature of the electrolyte; most studies are performed only with simple aqueous electrolytes, and mostly no comparative study in the different electrolytes is observed [63].

**Figure 2 gels-10-00438-f002:**
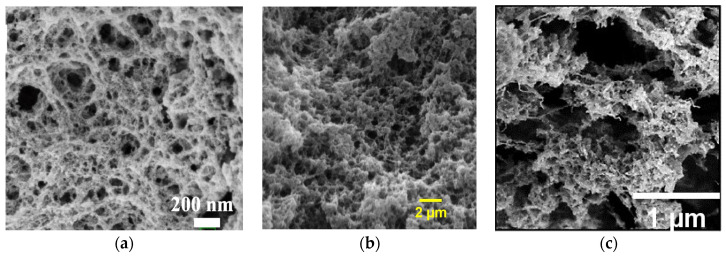
Polysaccharide-based bio-aerogels: SEM images of (**a**) lignin- [64], (**b**) cellulose-, and (**c**) chitosan- [4] aerogels. Reproduced with permission from (**a**) Wang, T., J. Chem. Eng.; published by Elsevier, 2023; (**c**) Chartier, C., Biomacromol.; published by ACS, 2023.

#### 2.3.4. Synthetic Counterparts

Nowadays, different synthetic counterparts can be used in the same applications as bio-based aerogels, and when talking about synthetic ones, organic electrodes are the most interesting. Therefore, different quinones, MXenes, and porous graphene-wrapped carbon foams are used [65]. Most developed electrodes for application in storage devices are carbon aerogels, usually prepared by the carbonization of resorcinol and formaldehyde [66]. Moreover, when it comes to separators, highly investigated ones could be divided into few groups—polymeric, ceramic, nonwovens, etc. However, the ones mostly used in LIBs are glass fibre and polypropylene separators [67].

## 3. Applications in Energy Storage—Batteries

### 3.1. Aerogels in LIBs

The high energy densities and extended lifetimes exhibited by lithium-ion batteries (LIBs) render them compelling candidates for energy storage exploration and advancement. Graphite, owning to its hierarchical structure, stands as a conventional anode material for LIBs. However, its limited intercalation capacity poses a significant drawback, propelling the quest for novel anode materials. While Li metal anodes present an ideal electrode by their electric properties, dendrite formation issues remain a challenge. Because of this and the pressing sustainability concerns, scientists have focused on carbon-derived anodes sourced from sustainable precursors, owing to their advantages in terms of cost-effectiveness, extended cycle life, and safety [68]. The predominant challenges regarding bio-based materials for active electrode use include low electrical conductivity, shrinkage during lithiation and delithiation processes, and uncontrolled lithium dendrite formation [69]. Improvement of electron transport in these materials could be achieved through pyrolysis, by accelerating the electrochemical reactions and fostering the formation of conductive carbon materials [70,71,72]. On the other hand, silicon-based materials have gained considerable attention as anodes primarily due to their high theoretical capacity (around 4200 mAh g^−1^), more than 10 times larger than graphite (around 370 mAh g^−1^). This higher theoretical capacity is crucial for the usage of electrical vehicles, as it facilitates power and energy density, while the battery size is reduced [73]. 

Although carbon-based materials can be readily derived from different bio-based sources, the challenge lies in disordered internal crystallites in hard carbon, i.e., in non-graphite carbon, leading to diminished reversibility and capacity loss. On the other hand, hard carbon is characterized as a highly stable material, so it is worth investigating. Factors such as additives, high porosity, and particle size significantly influence reversibility [74]. Decreasing particle size can notably reduce lithium-ion diffusion length, thus enhancing specific capacity and structural stability. By the pyrolysis of bio-based aerogels, different aerogels could be prepared, depending on the pyrolysis conditions. However, it is important for electrical and mechanical properties to optimize pyrolysis conditions in order to gain graphite-like structures, and not amorphous non-conductive carbon aerogels, as conductivity is a very important parameter for energy storage applications.

### 3.2. Bio-Based Aerogels as Electrodes in LIBs

Generally, carbon-based materials are currently regarded as practical anode materials for lithium-ion batteries, owing to their low cost and ease of preparation and manipulation regarding morphology and porosity. Most of the available scientific articles investigating lignin as the material for electrodes used in LIBs is based on cryogels or other porous materials. However, there is a limited number of research articles investigating lignin aerogels for this application. 

Besides pure lignin-based electrodes, different composite materials have been investigated. Huang et al. [75] prepared highly porous silica encapsulated in honeycomb-like porous carbon derived from lignin. Composite material like that showed an extremely high reversible capacity of 1109 mAh g^−1^, outstanding rate capability towards lithium storage, and great cycling performance. Their work can be seen in Figure 3. When talking about specific surface area and the porosity of their material, the authors provided information about SSA being remarkably high, 882 to 1107 m^2^ g^−1^, and confirmed that mesoporous structure is dominant in the materials. Such morphology leads to impedance being 130 ohm, meaning that the diffusion rate of lithium-ion is increased in the material with higher porosity. On the other hand, Zhou et al. investigated lignin-based HPMC material decorated with NiO nanoparticles, whereas they obtained highly graphitized carbon with a very high specific surface area (above 800 m^2^ g^−1^), with hierarchical pores, enhancing Li-ion transportation in the electrode, increasing the conductivity, and suppressing the deformation of NiO. More significantly, the hybrid nanosphere anode exhibited a high specific surface area of 852 m^2^ g^−1^, leading to a discharging capacity of 863 mAh g^−1^ at 0.1 A g^−1^ and was retained after 100 cycles for a LIB. Mesoporous structure was also confirmed. Importantly, Nyquist plots showed that the mechanism remains the same, only the value of impedance becomes slightly higher (from 50 to 55 ohms), after 1000 cycles [76]. The use of cellulose-based aerogel for the preparation of carbon-based anodes for LIBs provides a novel strategy for the synthesis of low-cost environmentally friendly LIBs with excellent rate performance and cycling. Li et al. showed that cellulose aerogel as a precursor for the carbon electrode preparation can result in a significant improvement of the cycling stability. The highly porous structure of the prepared material enhanced the stability of the formed SEI film, preventing electrodes from being damaged by the solvent. The authors obtained results indicating that aerogel properties, such as a high specific surface area of 621 m^2^ g^−1^, in comparison with the cellulosic precursor where the SSA measured was 53 m^2^ g^−1^, and high porosity, provided a large space for the storage of lithium ions. Moreover, lithium completely reacted at the electrolyte–electrode interface, further improving the stability of the electrode. Specifically, a high initial discharge capacity of 836.9 mAh g^−1^, a charge capacity of 803.3 mAh g^−1^, and an initial Coulombic efficiency of 96% were the characteristics of the cellulose-based anode material. The tenth cycle witnessed charge and discharge capacities of 579.3 and 589.3 mAh g^−1^, respectively. After 50 cycles, the porous carbon-based anode materials displayed a specific capacity of 543.5 mAh g^−1^ and a 99.37% retention rate. Lithium battery discharge mainly involved the alloying of lithium ions and the anode material, so the specific capacity reflected the capacity for the lithium-ion alloying process [77]. When browsing the literature, one can observe that most articles are based on cellulose I materials, and not many investigations are based on cellulose II, so oncoming investigations might be based more on cellulose II aerogels.

Li et al. [78] utilized marine biomass waste to synthesize N-doped porous carbon nanofiber aerogel via the pyrolysis of cellulose nanofibers in NH_3_, followed by activation with KOH. This carbon material boasts a high surface area of 2290 m^2^ g^−1^, facilitating increased contact between the active material and the electrolyte. Moreover, nitrogen doping enhances the conductivity of the electrode material, and like the authors showed, it also helped to increase SSA by up to 2290 m^2^ g^−1^. When evaluated as a lithium storage material, it demonstrates a remarkable specific capacity of 572 mAh g^−1^ at 1 A g^−1^ after 600 cycles and can be reversibly charged to 320, 263, or 220 mAh g^−1^ at the current density of 2, 5, and 10 A g^−1^, respectively [79]. Yang et al. [80] employed a microemulsion-templated sol–gel polymerization method to synthesize carbon aerogel. These aerogels possess a high surface area of 620 m^2^ g^−1^ and a 3D hierarchical porous structure. When utilized as anode materials in Li-ion batteries, they exhibit a reversible specific capacity of 470 mAh g^−1^ at 100 mA g^−1^, surpassing the theoretical capacity of 372 mAh g^−1^ for commercial graphite. The authors attributed this superior electrochemical performance to the developed carbon skeleton and continuous mesoporous structure, facilitating rapid transfer/diffusion of electrons and electrolyte ions, as well as a high specific surface area providing numerous active sites for Li^+^ storage reaction. Alex et al. [81], on the other hand, prepared resorcinol-formaldehyde-based carbon aerogel through ambient pressure drying, followed by carbonization at 1050 °C under an inert atmosphere. The resulting aerogel exhibited a high micropore volume fraction of 87%; investigated as an anode material with a high SSA of 534m^2^ g^−1^, for lithium-ion batteries, it demonstrated a specific charge capacity of 288 mAh g^−1^ and retained 97% of the initial capacity after 100 cycles. 

Ramirez et al. [82] synthesized cellulose-based carbon aerogels with uniformly distributed palladium domains throughout the carbonaceous matrix. Evaluation of the physicochemical properties revealed that favourable morphological characteristics of the carbon aerogels (high specific surface area—450 m^2^ g^−1^, hierarchical pore structure) were maintained upon the incorporation of palladium nanoparticles. This suggests that Pd particles do not hinder pores within the carbon framework, thus preserving efficient ionic transport in the hybrid carbon aerogels. The presence of oxygenated groups on the carbonaceous material surface is essential for the immobilization of the metallic domains within the investigated aerogels. The immobilization occurs via coordination bonds between the cellulose groups and the palladium, preventing the formation of rigid sheets of connected aromatic rings. Consequently, the crystalline properties of the carbon aerogel were modified by the addition of palladium, as Pd particles facilitated the reduction in graphite crystals and promoted structural disorder. Additionally, except for preserving the advantageous textural properties, Pd-loading significantly enhanced the electrical properties of the composite material, leading to remarkable improvements in specific capacitance, rate capability, and power densities. This enhancement in the global conductivity of the carbon network upon increasing Pd load allows for more efficient utilization of the electrochemically active surface area and a significant reduction in the charge diffusion resistance. Impedance characterisation showed that the mechanism and the pathway of electrons remains the same in all investigated samples, so the Nyquist plots are the same in terms of the shape, but impedance values are decreasing to values of approximately 50 ohms, in comparison to the pure cellulosic sample (being 200 ohms). These characteristics position this hybrid material as a promising candidate for energy storage applications.

Liu et al. performed DFT calculations based on cellulose-based carbon aerogels supported by rutile nanoparticles (Ru@CCA) to understand the catalytic mechanism and to show the structure–performance relationship between aerogel support and metal nanoparticles. The authors fabricated 3D supporting bio-based aerogel material modified with metal nanoparticles on the surface. Both DFT calculations and the following experimental results demonstrate that the interface of ruthenium–carbon in Ru@CCA can efficiently stimulate the kinetics of CO_2_ reduction and evolution reaction. When assembled in the lithium-carbonate battery, the battery reveals a significantly enhanced energy efficiency of 80%, a high discharge capacity of 10.7 mAh cm^−2^ at 20 μA cm^−2^, and a cycle life of 421 cycles at 100 μA cm^−2^. Consequently, the enhanced performance of Li–CO_2_ batteries with non-noble metals embedded in cellulose-based carbon aerogels (i.e., Co@CCA, Fe@CCA, and Ni@CCA) further confirms the viability of this cathode catalyst design strategy. Furthermore, the flexibility of the carbon aerogel electrode enables the device to bend and even fold, making it a great candidate for applications in wearable electronics [83]. Xu et al. showed how different bio-based materials could lead to advantageous properties, i.e., cellulose–lignin aerogel showed improved properties in comparison to pure aerogels from cellulose and lignin [84]. Except for very good specific capacitances (around 120 F g^−1^), tested aerogels as electrodes showed exceptional cycle stability, with 98% capacity retention after 10 thousand charge–discharge cycles. Nowadays, different influences have been investigated on the properties of carbon aerogels. Thus, Cao et al. investigated the influence of solvent on the gelation time and the following properties of prepared lignin–resorcinol–formaldehyde (LRF) aerogels for application in different energy storage devices. The authors prepared lignin nanoparticles and used green deep eutectic solvent, and the resulting lignin-derived carbon aerogel with a typical porous structure (SSA being 421 m^2^ g^−1^) was consequently prepared. The storage capacity of those aerogels was 284 F g^−1^ and cycle stability was 97% at 10 A g^−1^ over 5 thousand cycles, with remarkably low impedance where the series resistance measured was 0.45 ohms [85]. Table 1 represents the summarized results based on the bio-aerogels used in energy storage devices and their electrochemical performance. Additionally, different conducting fillers and multiple hetero atoms could be added into the matrix of non-conductive bio-based aerogels to enhance their conductivity, in addition to the pyrolysis process and carbonization. As an example, Bakierska et al. prepared a lithium titanate (Li_4_Ti_5_O_12_-LTO)/carbon aerogel, where potato starch was used as the carbon precursor, and where spinel LTO structure is preserved. The authors concluded that this material has excellent electrical and capacitance properties, along with stability [86].

### 3.3. Bio-Based Aerogels as Separators in LIBs

Within LIBs, a crucial component separating the anode and cathode is the separator, typically a porous polymeric membrane preventing electrical and physical contact between the electrodes while allowing ion transport [91]. Despite its passive nature, the separator’s characteristics, including porosity, pore size, thermal stability, high wettability, low shrinkage, and good mechanical strength, influence battery properties such as cycle life, ion transport efficiency, and overall performance and safety. Porous separators have to overcome various challenges within the electrochemical cell, usually affecting cell safety and performance. For instance, the charge/discharge process often triggers the oxidation of metal oxide cathodes and the reduction of anodes, leading to the decomposition of liquid electrolytes, blocking the pores and affecting the separators’ function [92]. Further, many charge/discharge processes usually lead to lithium dendrite formation, causing an internal short-circuit due to physical contact between the electrodes. The existence of different mistreatments during battery operation can physically damage the separators [93,94]. The culmination of all these factors can result in separator failure, ultimately leading to battery failure [95]. Commercialized lithium-ion batteries mostly use polyolefin micro-porous membranes due to their good stability (both mechanical and chemical). Many disadvantages in polyolefin membranes have to be overcome, such as poor thermal stability and low porosity of around 40%, causing insufficient electrolyte wettability. Furthermore, the low melting point of this type of separator can easily result in the shrinking of the separator and the internal short circuit. To address these issues and improve the thermal stability, electrolyte wettability, and porosity of the separator, numerous researchers have employed various materials and innovative techniques to prepare the membranes for Li-ion battery separators. Various approaches, including phase inversion, solution casting, surface modification, and electrospinning, have been proposed for the preparation of polymer membranes. Based on all characteristics that should be achieved in order to have a good separator material, high porosity of the aerogel materials, as well as high specific surface area, are the ones, which could lead to the formation of efficient separator materials. Nowadays, commercially used polyolefin substrates are modified with different ceramic nanoparticles, such as SiO_2_ [96] and Al_2_O_3_ [97] to improve properties such as ionic conductivity, thermal stability, wettability, etc. [98]. Besides the modification, different polymers have found their application as an efficient substitute for polyolefin separators [99,100,101]. Once again, those non-biodegradable and fossil fuel-based materials are not interesting due to environmental problems and complicated battery recycling [102]. 

As already explained, different non-biodegradable polymer materials have been investigated as a substitute for commercially used battery separators, but nowadays are mostly combined or totally replaced with bio-based materials. It is confirmed that higher porosity improves important properties of the separator material, making aerogels ideal materials for the investigation. Still, the presence of research papers on bio-based aerogel separators is scarce, but here will be presented some of the research works based on highly porous bio-based materials. Zhao et al. presented their work on lignin-modified polyacrylonitrile fibre membranes (L–PANs) prepared via the electrospinning method. The porosity of the prepared composites reached a maximum of 74% when the content of lignin was 50%. The results of the tested cycle performance indicated that the discharge capacity of the material containing 30 wt% of lignin could preserve 131.8 mAh g^−1^ after 50 cycles at 1C. On the other hand, the discharge capacity retention observed for that sample was 95% [103]. Highly porous carboxymethyl cellulose (CMC) membranes were investigated in terms of ionic conductivity and lithium-ion transfer. The observed values were successfully compared to commercial separators. In addition, it provides high thermal stability because the liquid electrolytes are well preserved in the polymer matrix of the carboxymethyl cellulose (CMC) due to the presence of –OHs and –COOHs, which significantly enhances the safety of the lithium-ion battery. When a porous membrane of CMC is used in the evaluation of LiFePO_4_, the cathode of the LiFePO_4_ provides high discharge capacity and good capacity retention, as well as good rate behaviour [104]. 

Gou et al., on the other hand, presented cellulose-based separators with properties much more similar to aerogels. The porosity of their material was up to 90%, and the pore size distribution showed that the size of the pores was in the meso-range. They adjusted the diameter of cellulose fibres, as well as the amount of polystyrene spheres. Through numerical simulations of the electrical kinetics and the distribution of lithium-ion concentration, it has been proved that the basic mechanism for lithium dendrite growth is a relationship between pore size distribution and the porosity of the sample. The separator with an excessive porosity showed defective performance at high current density, which was further characterized by the electrochemical measurements carried out on lithium-ion batteries. Moreover, the separators prepared by the authors were compared with other cellulose-based separators, and it was shown that they had better ionic conductivity than most investigated cellulose non-aerogel materials, as well as better mechanical properties. The authors performed the impedance investigation in terms of Nyquist plots, where Li/separator/Li cells were measured, and the obtained interfacial resistances were 167.5 ohms before and 262.1 ohms after cycling, indicating homogeneous Li^+^ deposition [105]. The electrospinning of polyimide nanofibers–cellulose composites was also investigated as lithium-ion battery separators. Hydrogen bonds incorporated into the sample enhanced both the flexibility and the mechanical strength of the material. The wettability of as-prepared separators was also tested and showed better properties than commercial polypropylene and polyimide separators, which significantly decreased impedance on the boundary of the separator and the electrolyte. By the electrical characterisation of prepared samples, the authors showed that the separator they prepared has the lowest resistance in comparison to the commercially used separators, PP and PI, respectively, being 159 ohms in the Li/separator/Li cell. Using cellulose–polyimide separators in LIBs leads to gaining an initial discharge capacity of 166.2 mAh g^−1^, with a capacity retention rate of 90% [106]. 

Wan et al., 2017, presented the preparation of cellulose-based aerogel membranes by a green route from cellulose solutions with ionic liquid. The yield porosities were higher than Celgard2400 (a commercially used separator, which has porosity of 40%). Moreover, the pore sizes were much lower than in the commercial separators, from 10 to 30 nm. The specific surface area of the obtained materials was 282 m^2^ g^−1^. Materials with pore sizes between 20 and 30 nm showed the highest values for electrolyte uptake and ionic conductivity, much higher than for the Celgard2400. For the sample with 95% porosity and pore size of ~20 nm, the electrolyte uptake was over 600%. More importantly, the prepared aerogels in the LIBs showed superior thermal resistance, meaning that batteries were stable, even at a temperature of 120 °C. By electrochemical investigation, the authors showed that their material gained much lower impedances than commercially used Celgard 2400—interfacial resistance being 360 ohms, lower than the 460 ohms of Celgard 2400 [55]. The work of Raafat et al. evaluates a cellulose–nanofiber aerogel (CNF-AG) separator that is highly flexible and environmentally benign. It analyses the separator’s dynamic behaviour in battery cells. The performance is better than the commercially used glass fibres due to the customized channel-like structure’s high porosity of 99.5% and strong mechanical stability. Its well-connected pore network and affinity for carbonate-based electrolytes allow it to absorb 12,000% of the electrolyte. Moreover, an ionic conductivity of 2.64 mS cm^−1^ was obtained, with an effective diffusion coefficient of 1.70 × 10^−10^ m^2^ s^−1^, only 16% lower than that of the bulk electrolyte. Excellent electrochemical performance is noted, with good cycling stability reaching 200 cycles. The exceptional structure–performance linkages of the CNF-AG demonstrate its superiority as a biodegradable separator that conforms to shape and is appropriate for use in metal-ion batteries [107]. Moreover, after presenting some of the materials which are not pure aerogels (modified aerogels, or aerogel-modified commercially used separators), there is interesting work on how squid-pen could be used as the source of chitin for making efficient chitin aerogels [108]. The authors prepared chitin-based aerogel with a 3D nanofibrous skeleton via a freeze-drying method. The resultant chitin nanofibrous aerogel (ChNFA) was flexible, showing good thermal stability and a high specific surface area of 275 m^2^ g^−1^. The already mentioned properties are enough for testing the prepared material as the candidate for substitution of commercially used separators. When compared to Cellgard2400, chitin-based aerogel exhibited a higher electrolyte uptake of 3278%, as well as ionic conductivity (being 1.8 mS cm^−1^, respectively). As expected, chitin-based aerogel showed an enhanced rate performance of 92.2 mAh g^−1^ at 20C, with a cycle life of 98.2% after 500 cycles. 

What is also the focus of the investigation nowadays is the modification of commercialized porous structures with aerogels to enhance some of the properties. Therefore, Liao et al. successfully coated cellulose aerogel onto a commercialized polypropylene separator with a macroporous structure. The prepared sample had better porosity properties and therefore better wettability in general and, consequently, a higher ionic conductivity than the pure polypropylene sample. As expected, this led to an improvement in the battery performance, especially the cycling performance and the rate behaviour [109]. Zhu et al., on the other hand, prepared a composite separator consisting of cellulose fibres and polyphenylene sulphide. In comparison with commercial Celgard 2400, their separator was more porous, and therefore the electrolyte uptake ability was improved (260%). Improved uptake leads to improved ionic conductivity and lower interfacial resistance with lithium, and thus better electrochemical properties. The LIB assembled with the prepared separator shows stable cycle performance and good C-rate capability, with a discharge capacity retention rate of 90% after a hundred cycles [110]. Even though the authors did not use aerogels as materials for the investigation, one can see that an increase in porosity is followed by an increase in the electrolyte uptake and better electrochemical properties; this conclusion can lead to the investigation of cellulose aerogels as promising separator materials in LIBs. He et al. presented in their work the development of a non-aerogel mesoporous membrane derived from cellulose and metal–organic frameworks (MOFs) with the coordination effect. When used as a separator for LIBs, the battery exhibits 153 mAh g^−1^, with a capacity retention of 98% after two hundred cycles [111]. Figure 4 presents the morphological and electrochemical properties of the selected separators.

### 3.4. Bio-Based Aerogels in Other Batteries

#### 3.4.1. Bio-Based Aerogels in Lithium–Sulphur Batteries

Lithium-ion batteries are predominantly investigated, but some of the research papers are based on lithium–sulphur batteries and their performance. Specifically, one of the latest studies was devoted to the preparation of a hard framework consisting of 1D TEMPO-oxidized cellulose nanofibers (TEMPO-CNF) modified with 2D rGO. Afterwards, the prepared framework was uniformly embedded into the monolayer of MXene. Composite 3D aerogel could serve as a hybrid sulphur host material, minimizing the effect of sulphur volume expansion during charge/discharge cycles. The prepared materials were tested as cathode materials in lithium–sulphur batteries and showed great improvement in rate capability, i.e., it was 744 mAh g^−1^ at 5C, along with internal resistance being 2.2 ohms. Moreover, electrodes like this showed a high capacity of 869 after two hundred cycles, with a low capacity-fading rate [87]. Previously, bacterial cellulose was used as the template for the preparation of cathode material with high sulphur content, making the flexible 3D aerogel material a carbon support for sulphur. The unique interconnected networks of bacterial–cellulose-based carbon aerogel enabled the composite to have good electrical conductivity as well as a robust framework to withstand the strain caused by the volume changes of active materials. The macroporous structure of the composite allowed for the rapid movement of lithium ions in the organic electrolyte. The ultralight BC interlayer inserted between the sulphur cathode and the separator could also reduce the total electrode resistance. Most importantly, the CBC interlayer was able to effectively relieve the excess accumulation of sulphur on the cathode surface. Migrating polysulphides that usually settle on the cathode surface now land on the new CBC interlayer. The combined effects of this configuration of CBC on the cathode side are a low-cost, eco-friendly electrode material design for the future practical application of high-performance lithium–sulphur batteries [112]. Bio-based aerogels could be derived directly from natural sources, so Zhu et al. showed that carbon-based aerogels prepared using sweet potato as the source could be used as the efficient separator material of lithium–sulphur batteries. Specifically, the prepared carbon-aerogel served as the modifier of the commercially used separator so properties such as poor cycle life and low utilization rate could be improved. Coated carbon aerogel reduced the battery resistance, but in addition, it served as an upper current collector by increasing the utilization rate of the sulphur. Moreover, the initial discharge capacity was much higher than in the pure commercial separators, being 1216 mAh g^−1^ at 0.1 C, followed by the reversible discharge capacity of 431 mAh g^−1^ after one thousand cycles at 1C, where Coulombic efficiency was over 95% [113].

#### 3.4.2. Bio-Based Aerogels in Zinc Batteries

Fu et al. proposed a high-strength bamboo cellulose-based membrane (BCM), uniquely designed for zinc-based batteries. BCM exhibits a hierarchical structure, with interlinked cellulose nanofibers and formed 2D nanosheets with natural nano-scale pores. This structure, governed by van der Waals forces and H-bonding, provides exceptional mechanical strength, effectively preventing zinc dendrite formation compared to glass fibre (GF) separators. Furthermore, BCM maintains structural integrity, even after immersion in neutral and alkaline electrolytes, while GF is easily damaged. The mesopores in BCM facilitate uniform Zn^2+^ flux for homogeneous deposition. Consequently, Zn/BCM/Zn symmetric cells demonstrate an extended cycle life of up to 5000 h, while the Zn/GF/Zn symmetric cell only displays 30 h. BCM, derived from bamboo, a renewable and abundant biomass, offers a green and pollution-free preparation process [114]. 

Figure 5 presents selected results based on electrochemical and morphological results carried out on bio-aerogels as the main parts of the battery cell. Figure 5a,b present the already explained results from the literature, and in Figure 5c we wanted to present our results to emphasize the possibility of using cellulose-based aerogels as separator material in LIBs. GDC curves have additional contributions in comparison with GF, which will be additionally worked on, but the important thing is that there are many parameters that should and will be worked on in order to improve cellulose-aerogels as viable substitutes for commercially used non-biodegradable materials.

## 4. Applications in Energy Storage—Supercapacitors

It is well-established that ideal electrode materials should feature hierarchical porous structures encompassing micropores, mesopores, and macropores. Macropores (>50 nm), functioning as large ion buffer storage tanks, can significantly reduce ion diffusion distances. Mesopores (2~50 nm) offer low-resistance pathways for rapid ion transport to the internal areas of the electrode, and micropores (<2 nm) provide numerous ion storage active sites and multiple ion diffusion channels. Carbon aerogels are frequently utilized in the fabrication of electrodes for supercapacitors due to their unique three-dimensional network structure, excellent conductivity, and cost-effectiveness. [64,115,116]. Xu et al. showed how lignin replacement of resorcinol could lead to the formation of a very efficient electrode material. The authors prepared lignin-based carbon aerogel by the sol–gel polymerization of lignin (L) and resorcinol (R) with formaldehyde (F), while the resorcinol was partially replaced with lignin in different mass ratios. The replacement of 20 wt% was proven to be the sample with the best morphological and electrical properties. The specific surface area gained was 590 m^2^ g^−1^, and when talking about electrical properties the specific capacitance of the prepared electrode reached 142.8 F g^−1^ at a current density of 0.5 A g^−1^. Even with the increase of the current density (up to 10 A g^−1^), the specific capacitance of the prepared aerogel electrode remained at 112.5 F g^−1^. The observed values indicate an ideal capacitive behaviour followed by a good rate capability of the sample. The lignin-based carbon aerogel showed excellent durability, with the specific capacitance remaining at 96% after two thousand cycles, which is related to the interconnected porous network structure of the prepared aerogel, in comparison to samples without lignin [88]. 

A symmetric supercapacitor (SSC) fabricated with two carbon aerogel electrodes exhibited remarkable performance, with a specific capacitance of 193 F g^−1^ at 0.5 A g^−1^ and a maximum energy density of 28.2 Wh kg^−1^ at a power density of 262.6 W kg^−1^ [85]. The inclusion of crosslinker content in the formation of lignin-based aerogels plays a pivotal role in crafting the unique and controllable porous structure of carbon aerogels. In that manner, Karaaslan et al. utilized formaldehyde as a crosslinker and demonstrated that the lignin-to-crosslinker ratio significantly influences various properties of the prepared aerogels, particularly their suitability for applications such as supercapacitor electrodes. They found that a higher lignin content during the formation of carbon aerogels leads to the formation of better ion accessibility [117]. Superior electrochemical performance and cyclic stability were observed in comparison to commercial activated carbon and RF-based carbon aerogel tested under the same two-electrode system. Additionally, surface activation further enhanced the performance of electric double-layer capacitors (EDLCs), improving specific capacitance and capacitive rate retention due to modifications in the internal pore structure of carbon aerogels. Symmetric supercapacitor cells assembled with activated carbon aerogel containing 87 wt% lignin exhibited an impressive energy density of 3.2 Wh kg^−1^ at a power density of 209.1 W kg^−1^. These results highlight the potential of lignin–epoxy-based activated carbon aerogels as promising, cost-effective, and renewable materials for various energy applications, including supercapacitors and carbon electrodes for other hybrid battery systems such as Li-ion capacitors [118].

Hierarchically porous carbon aerogels were synthesized through a green process involving ice-templating, lyophilization, and subsequent carbonization. The authors claim that they, for the first time, reported carbon aerogels prepared at a cooling rate of 7.5 K min^−1^, achieving an exceptionally high specific surface area of 1260 m^2^ g^−1^ without any physical or chemical activation. The electrode made from this aerogel exhibited outstanding electrochemical performance, with a specific capacitance of 410 F g^−1^ at 2 mV s^−1^ and a cyclic stability of 94% after 4500 charge and discharge cycles. The impact of the ice-templating cooling rate and the solid content of lignin and cellulose nanofibers in the suspension on the structure and electrochemical performance of the carbon aerogels was studied. It was found that the ice-templating process and cooling rate significantly influenced the nanoporous structure and SSA of the carbon aerogels, whereas the solid content of the lignin–nanocellulose suspension had minimal effects. Assembled as a supercapacitor, the CA electrode achieved a specific capacitance of 240 F g^−1^ at 0.1 A g^−1^. The SC exhibited a relaxation time constant of 1.3 s, indicating a fast response, and an energy density of 4.3 Wh kg^−1^ at a power density of 500 W kg^−1^. This study, therefore, paves the way for the development of green, high-performance, environmentally friendly carbon aerogel electrodes for energy storage applications [89]. Dong et al. showed the excellent properties of the bidirectional freezing process for the preparation of a composite aerogel consisting of cellulose nanofibrils and graphene oxide. By the temperature gradient produced with bidirectional freezing, the uniformly dispersed composite suspensions are repulsed by the development of layered ice crystals. By carbonization, carbon aerogels with regular layered structures are formed. When talking about the mechanical properties of carbon aerogels prepared by a bidirectional freezing process, they showed extreme mechanical properties and very good electrochemical performance. The supercapacitor electrodes derived from these aerogels displayed a high specific capacity of 140 F g^−1^ at 0.1 A g^−1^, with a capacitance retention of 93% after 5000 cycles at 2 A g^−1^ [90]. Haj et al. developed an advanced cellulose nanocrystal-based carbon aerogel for adaptive supercapacitor performance using different salt anions, chlorides, citrates, and acetates. The authors showed that, at a current density of 1 A g^−1^, the carbon aerogel (CA) with nitrates had higher specific capacitance in comparison with carbon aerogels with other salts. The specific capacitance of the nitrate–CA electrode at a current density of 1 A g^−1^ was 413.3 F g^−1^, and their work can be seen in Figure 6b,c [119]. 

Nowadays, special attention is devoted to graphene-based or graphene-modified materials, due to their significant properties in terms of electrical conductivity and capacitance. Mainly, the problem for graphene-based materials is in the layer-stacking aggregation, so for efficient applications it should be stabilised with different materials. Thus, Wang et al. [120] presented their synthesis and potential applications of TEMPO-oxidised nanocellulose/pristine graphene aerogels, which were investigated as electrode materials in supercapacitors. The authors investigated how the carbonisation temperature influenced the properties of the material and concluded that by an increase of temperature to 1100 °C the electrochemical performance of the electrode improved noticeably, i.e., specific capacitance gained was 361.74 F g^−1^ at a current density of 0.5 A g^−1^ and the capacitance retention was 99.3% after 5000 cycles. Using EIS, the authors investigated the interface of the contacts and concluded that impedance decreased to 4 ohms for the sample showing the best supercapacitor performance, indicating best interface adhesion of the constituents.

Xiao et al. [121], on the other hand, carried out the investigation based on cellulose–aerogel-derived porous biochar via heteroatomic doping and investigated electrode supercapacitor performance. The authors presented the synthesis of N,B-co-doped porous biochar (NBCPB) with temperature activation at 650 °C, and they concluded that the cellulose (or, for that matter, carbonic) structure is not influenced by the NBCPB preparation, which was confirmed by the XRD analysis. The prepared samples gained high specific surface areas from 891 m^2^ g^−1^, with the minimum average pore size of 6.6 nm. As expected from these properties, the specific capacitance of the samples were high (220.9 F g^−1^ at 1 A g^−1^), as was the capacitance retention—98.6% after 3000 cycles. Via the electrochemical investigation, the authors confirmed low resistance values due to the nitrogen and boron doping, in addition to well-connected structures within the NBCPB framework. In Figure 6, selected samples and their morphological and electrochemical performance when used in supercapacitors are presented. The authors showed successful synthesis of 3D hierarchical porous and conductive carbon aerogel material derived from CNF/rGO/Zn/Co ZIF (CRZC) via an innovative and simple process involving directional freeze drying and high-temperature carbonization. The synergy of multiple materials and the unique 3D hierarchical porous structure, featuring micropores, mesopores, and macropores, enabled the CRZC electrode to achieve a high specific capacitance (364.6 F g^−1^ at 1 A g^−1^) and an excellent capacitance retention rate (83.4% at 10 A g^−1^). Additionally, the symmetric supercapacitor (SC) assembled with CRZ demonstrated a maximum energy density of 18.9 Wh kg^−1^ at 288.4 W kg^−1^, showcasing its broad potential in energy storage. Notably, it also exhibited remarkable cyclic performance with 78.9% capacitance retention over 10 thousand cycles. Consequently, the CRZC hybrid carbon aerogel shows significant promise as an electrode material for SCs [122]. In Table 2 are summarized the properties of bio-based aerogel separators and energy storage device properties when those separators are used.

## 5. Applications in Fuel Cells

Many disadvantages provided by powders in terms of catalysts could be solved by using monolithic materials. For that matter, different polymer aerogels with high specific surface area could be used, enabling hydrogen diffusion and transport. Ideally, an inorganic matrix with acceptable mechanical properties could be combined with different bio-based materials with other properties, except mechanical. Therefore, Ye et al. showed that using bio-based materials could significantly simplify H_2_ generation systems. They showed that the hydrolysis of NaBH_4_ to H_2_ regulated simply by adjusting the position of the catalyst, enabling on-demand production. Moreover, the used aerogel catalyst can be cleaned through repeated compression and reused in the aqueous solution, retaining its 3D porous structure and making it easily recyclable [123]. Nowadays, except for energy storage systems, great focus is being devoted to harvesting energy from wastewater. Thus, Yang et al. [124] investigated chitosan aerogels as efficient supports for an Fe–N–C catalyst for microbial fuel cells. Chitosan was chosen as the N-containing precursor, leading to the increase of the power density, proving to be an economically viable solution in microbial fuel cells. The addition of the filling polymer matrix and combining it with cellulose aerogel was investigated by Lan et al. as the electrodes for fuel cells [125]. The sample consisting of quaternized branched polyethyleneimine linked with cellulose-aerogel showing the best chemical stability and the high ionic conductivity was tested as the constituent of the fuel cell. The prepared anion exchange membrane in the fuel cell application led to a maximum power density of 46.3 mW cm^−2^, being quite low, but good enough for further investigation. The electrochemical investigation provided the results of the energy of activation being around 16 kJ mol^−1^, confirming the ionic conductivity of the system.

In the main, bio-based aerogels are investigated for applications in batteries and supercapacitors, so for this reason, this section could be discussed in some other review paper separately.

## 6. Conclusions Regarding Future and Challenges

Bio-based aerogels are viable materials for several segments of energy storage systems, such as rechargeable batteries, supercapacitors, and fuel cells. All reports agree that the combination of their recognizable properties, mesoporosity, high specific surface area, biocompatibility, and biodegradability, will continue to ensure their presence as constituents of the mentioned energy storage systems. Therein, the bio-aerogels primarily serve as electrodes and separators, with their high specific surface area, porosity, and conductivity, thereby derivatively offering better energy and power density and a longer lifecycle of devices.

In comparison to conventional aerogels, bio-aerogels will continue to be more suitable for a range of respective energy storage applications, primarily due to fulfilled sustainability and environmental considerations. When commenting on the properties of electrodes based on bio-aerogels, the presented values of specific densities are gaining 1109 mAh g^−1^ of the battery cell, and when talking about separator properties, charge–discharge capacities are in a similar range to commercially used non-biodegradable separators, being in the range of 150 mAh g^−1^. The observed values are presented in Table 1 and Table 2, and the impedance properties are presented in Figure 4. The challenges will continue to be related to broadening the threshold of renewable sources beyond cellulose, lignin, and chitosan, and advancing the synthesis (wet-chemistry routes) and processing (drying) routes to make the processing broadly scalable (both in the scale-up and in the scale-down directions).

At this point, a research gap is visible in the methodological inability to prepare adequate quantities of specific functional aerogels in a standardized manner. Therefore, the potential future opportunities surely lie in the domain of bio-aerogel composites that will enable miniaturization on behalf of better energy densities and more widespread production, addressing the cost-efficiency aspect, in addition to the already mentioned positive environmental aspects. Bio-aerogels surely display a positively transformative potential beyond energy storage systems.

## Figures and Tables

**Figure 1 gels-10-00438-f001:**
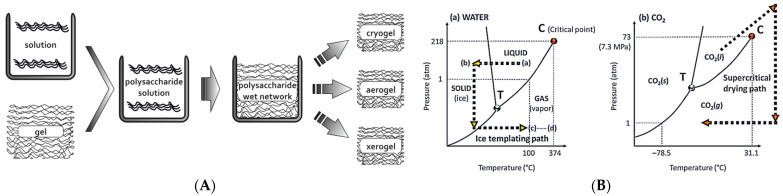
Aerogels: (**A**) Solution-chemistry synthesis shown schematically; (**B**) Drying processing.

**Figure 3 gels-10-00438-f003:**
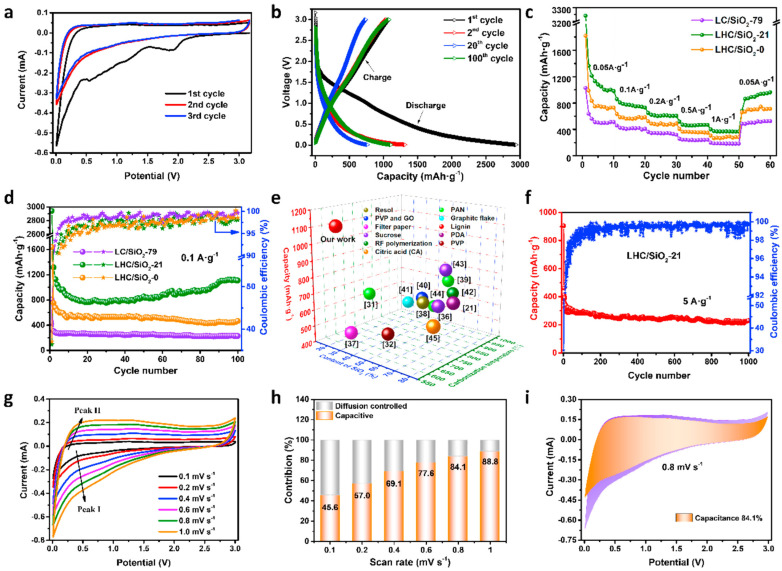
Bio-aerogel electrodes [75]. Testing of lignin–SiO_2_ aerogels as an electrode in LIB: (**a**) CV curves at a scan rate of 0.1 mV, where different peaks can be observed. Two irreversible peaks during the first discharge process—corresponding to the formation of SEI layer and irreversible reactions between SiO_2_ and Li+ ions; (**b**) charge/discharge curves at a current of 0.1 A g^−1^, where similar behaviour as on CV curves could be observed; (**c**) rate properties at different current densities; (**d**) Coulombic efficiency and cycling performance at a current of 0.1 A g^−1^ of different lignin–SiO_2_ samples; (**e**) capacities compared between high-capacity materials reported in the literature and the lignin–SiO_2_ from this research paper; (**f**) long-term cycling stabilities at a high current density of 5 A g^−1^; (**g**) CV curves at different scan rates; (**h**) normalized contribution ratio of capacitive at different scan rates, where the capacitive contribution slowly increases with the scan rate increase, indicating larger capacitance contribution favouring Li^+^ storage; (**i**) the contribution ratio of pseudocapacitance at 0.8 mV s^−1^, by which good charge transfer kinetics is confirmed. Reproduced with permission from Huang, S., Micropor. Mesopor. Mater.; published by Elsevier, 2021.

**Figure 4 gels-10-00438-f004:**
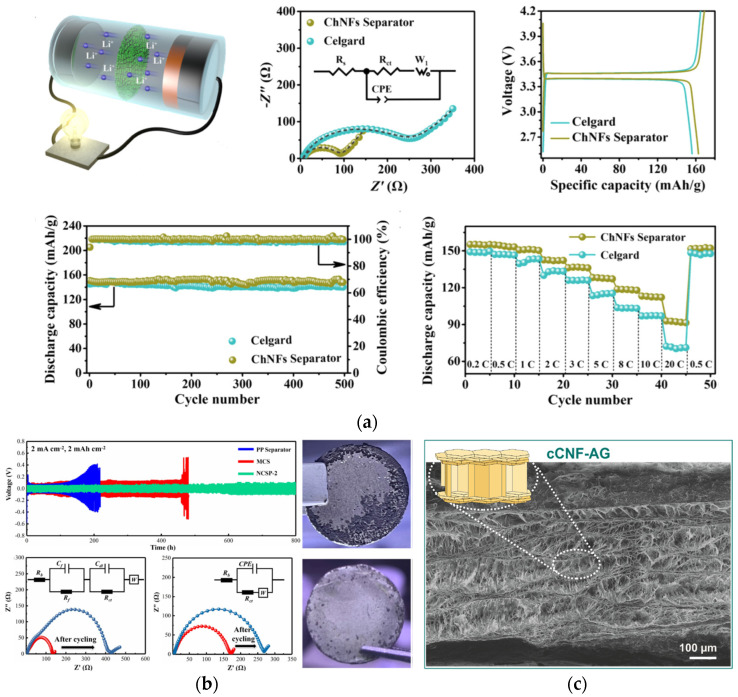
Bio-aerogel separators: (**a**) Battery performances of half cells assembled with chitosan NF separators; schematic illustration of the half cells; Nyquist plots—indicating that the interfacial resistance of the Celgard separators is three times higher than the chitosan ones; initial charge–discharge profiles—cells with chitosan separators have an initial discharge capacity of 163 mAh g^−1^, being slightly higher than the one with the Celgard separator; cycling performance is excellent in terms of cyclability; rate capability of LiFePO_4_ half cells, where the chitosan separator shows higher discharge capacity in the whole range of current densities [108]. (**b**) Temporal voltage curves of the cells assembled with the investigated separators at a current density of 2 mA cm^−2^ for 1 h per cycle are visible. For the Li/microcellulose/Li cell, the voltage is ~100 mV, with a slight increase to ≈150 mV at 114 h, and stable polarization is visible for ≈330 h, after which voltage dramatically rises to ≈570 mV at 482 h, suggesting that the cell fails in the regulation of the Li deposition. The cell using the nanocellulose has a consistent time-dependent voltage curve, being below 140 mV, displaying a uniform Li^+^ electrodepositing procedure. By the Nyquist plots, a noticeable variation of the interfacial resistance and an unstable interface is detected when using the microcellulose separator, but while using the nanocellulose separator, homogeneous Li^+^ deposition is observed. A rough surface with a noteworthy amount of dendritic structures is observed on the lithium foil cycled with the microcellulose, while the electrode cycled with nanocellulose exhibits a relatively smooth and uniform surface [105]. (**c**) SEM images of the longitudinal section of the compressed cellulose–nanofiber aerogel (CNF-AG). The schematic illustration of CNF-AG displays the densified areas and preserved channel-like domains [107]. Reproduced with permission from Yang, X., J. Power Sources; published by Elsevier, 2023; from Gou, J., J. Membr. Sci.; published by Elsevier, 2022; from Raafat, L., ACS Appl. Energy Mater.; published by ACS, 2020.

**Figure 5 gels-10-00438-f005:**
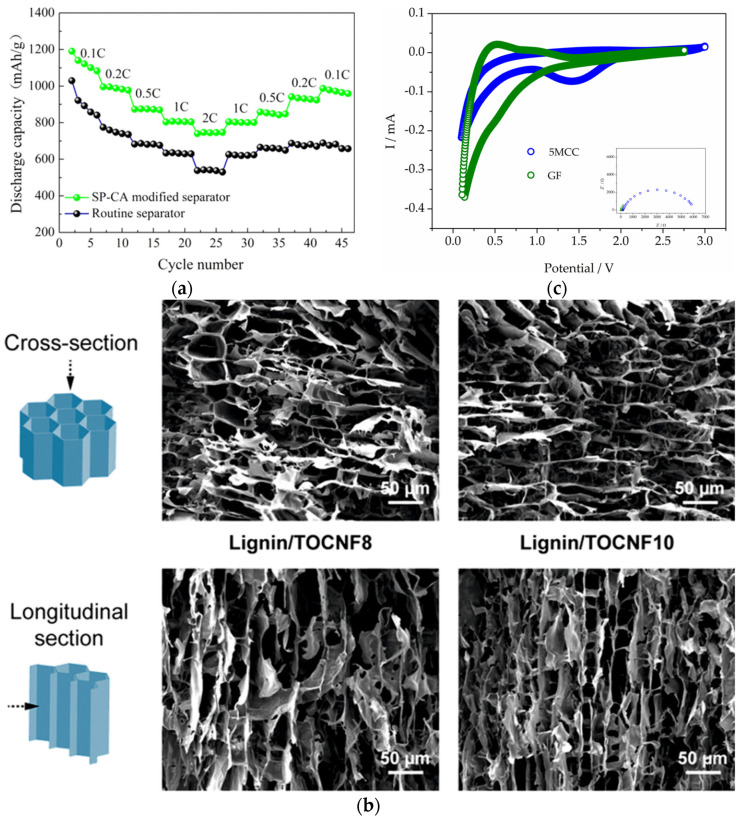
Selected results and our results: (**a**) The rate capability of the Li–S cell with a routine separator (commercially used one) and sweet potato-based aerogel-modified separator demonstrated the improvement of the rate capability with the bio-based separator, which could be correlated to the highly porous structure of the aerogel helping the uniform distribution of sulphur species. Also, the porous structure of carbon aerogel improves the electrolyte uptake, which makes the migration of lithium ions easier [113]. (**b**) SEM images of the cross-section and longitudinal section of the lignin/TEMPO-oxidized cellulose nanofibers (TOCNF) precursors with different TOCNF contents. An anisotropic macroporous structure produced by the ice-templating process is clearly visible [47]. (**c**) CV and Nyquist plots observed using microcrystalline cellulose aerogels as separator materials in LIBs. Reproduced with permission from Zhu, L., ACS Sustain Chem. Eng.; published by ACS, 2018; from Geng, S., ACS Appl. Mater. Interfac.; published by ACS, 2020.

**Figure 6 gels-10-00438-f006:**
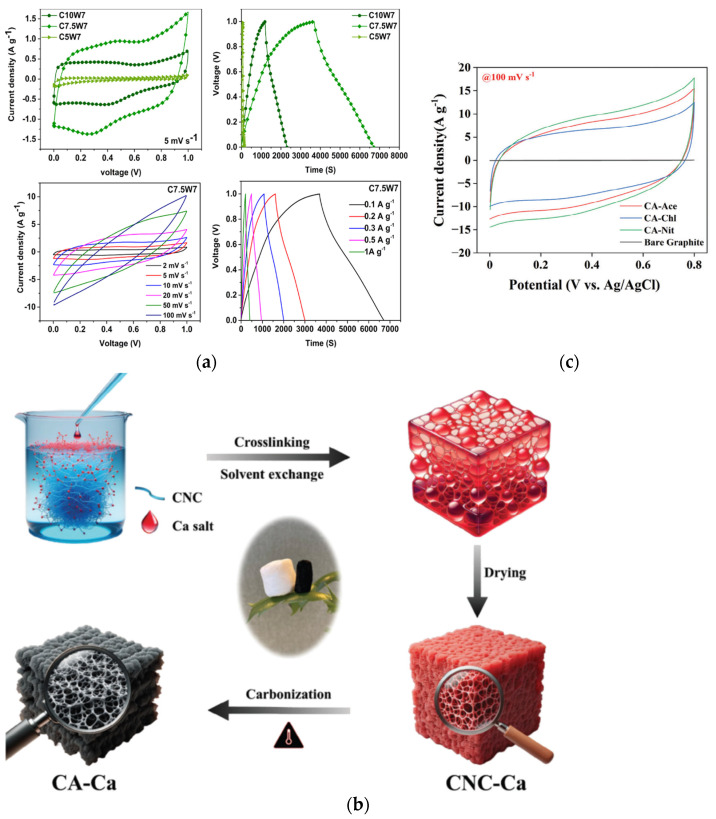
(**a**) Electrochemical properties of carbon aerogels measured using a three-electrode cell with 1 M H_2_SO_4_ as the electrolyte. First, CV and GCD measurements represent that the highest electrochemical performance was observed for the sample having a specific capacitance of 410 F g^−1^ at 2 mV s^−1^ and 303.5 F g^−1^ at 0.1 A g^−1^. CVs are near-rectangular-shaped at lower scan rates, and the deviation to a quasi-rectangular shape toward the higher scan rates is also represented. CV and GCD curves for a chosen sample at different scan rates and current densities are shown. It can be seen that by the decrease in the cooling rate (from 10 to 7.5 °C min^−1^), electrochemical properties are highly improved, probably due to the improvement in SSA values [89]. (**b**) Schematic illustration of the preparation of cross-linked carbon aerogels [119]. (**c**) CV curves of different aerogels, i.e., different salts used in the crosslinking procedure at a constant scan rate of 100 mV s^−1^, with an operating potential of 0–0.8 V. The CV curve for bare graphite displays the lowest current density in comparison to those of the carbon aerogel-based electrodes, demonstrating that the active materials’ capacitive performance remains unaffected under the same electrochemical conditions. The electrode prepared from nitrate-cross-linked carbon aerogel displays a larger voltammogram area compared to those of acetate and citrate-based ones, suggesting an improvement in the electrochemical performance. Improvement in electrochemical performance is followed by higher specific surface area values [119]. Reproduced with permission from Thomas, B., ACS Appl. Nano Mater.; published by ACS, 2022; from Al Haj, Y., Adv. Funct. Mater.; published by Wiley, 2024.

**Table 1 gels-10-00438-t001:** Summarized performance of bio-aerogels as electrodes in energy-storage devices.

Material	Specific Capacity/mAh g^−1^	Coulombic Efficiency/%	Number of Cycles	Device	Ref.
lignin	1109	99	100	LIB	[75]
lignin	863	92	100	LIB	[76]
cellulose	570	99.97	100	LIB	[77]
cellulose	572	/	600	LIB	[78]
resorcinol-formaldehyde	288	97	100	LIB	[81]
Pd/cellulose	200 *	93	1000	LIB	[82]
Lignin–resorcinol–formaldehyde	193 *	97	5000	LIB	[85]
TEMPO-cellulose	869	99.7	200	LSB	[87]
lignin	112.5 *	96	2000	supercapacitor	[88]
lignin/CNF	410 *	94	4500	supercapacitor	[89]
cellulose	140 *	93	5000	supercapacitor	[90]

* F g^−1^, referring to capacitance.

**Table 2 gels-10-00438-t002:** Summarized performance of bio-aerogels as separators in energy storage systems.

Material	Porosity/%	Pore Size/nm	Charge/Discharge Capacity/mAh g^−1^	Number of Cycles	Ref.
PP	42	/	137.5	-	[103]
Celgard 2400	41	/	138.2	100	[55]
Celgard 2730	/	/	126.30	1000	[104]
lignin	74	/	131.8	50	[103]
cellulose	62.5	50–200	140.0	40	[104]
cellulose	61.5	100	143.6	400	[105]
cellulose/polyimide	78	/	166.2	200	[106]
cellulose	95	10–30	138.1	100	[55]
chitosan	98.4	40–60	92.2	5000	[108]
cellulose/PP	73	/	150.0	100	[109]

## Data Availability

Not applicable.

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
