# Peer review of "Bio-Based Aerogels in Energy Storage Systems"

_gels, 2024, doi:10.3390/gels10070438_

Round 1

Reviewer 1 Report

Comments and Suggestions for Authors

Supercapacitors are attractive energy storage devices, characterized by rapid charge-discharge, long cyclability with high power densities. Various transition metal oxides and hydroxides have been researched as potential electrodes for supercapacitors due to their excellent specific capacitance derived from faradaic/electrostatic reactions on the surface. In the submitted work, the authors have reviewed aerogels having low density nanoporous solids having open fine structures for various energy storage applications including batteries, SCs and fuel cells. The reported review focused on the bio-based aerogel; synthesis, properties and other challenges seen in the applications. This is a significant area of research; however, the work requires lot of clarity. The review in its present form is not suitable and needs some revisions before rendering a final decision.

My specific points are below

·         Graphene based aerogels are getting wide attention in the area of SCs.

·         The cost effectiveness of the fabricated supercapacitor can be addressed by cheaper electrode preparation and processing methods using low-cost carbon precursors as source materials

·         The conductivity of the aerogels can be improved by doping with conducting fillers and multiple hetero atoms – which must be included in the appropriate sections.

·         Carbon aerogels derived from these resorcinol formaldehyde-aerogels have a small mesopore surface area, however an especially large micropore area. They provide electrical capacities which are most suitable for their use in supercapacitors.

·         What is “solid-state impedance spectroscopy”?

·   Materials like lignin, chitosan, alginate, cellulose for electrodes in supercapacitors and battery applications. This could be included and discussed in the synthesis and applications sections of the review article.

·         Authors have confused between battery (mAh/g) and SC (F/g), in the table it must be clearly identified.

·         References are not correctly embedded in the text. One simple example, Ramirez et al {no reference}.

·         Figure 6. (a) immediately the reference number [115] – this looks weird.

·         The last application (Fuel Cells) is written very weak.

·         Section 6 must include some performance metrics.

·         Section 7 – is it essential? Why materials and methods for review paper? Unless otherwise, authors have included their own original work.

Comments on the Quality of English Language

Some edits are required. Figure captions are too lengthy.

Author Response

REVIEWER 1

Supercapacitors are attractive energy storage devices, characterized by rapid charge-discharge, long cyclability with high power densities. Various transition metal oxides and hydroxides have been researched as potential electrodes for supercapacitors due to their excellent specific capacitance derived from faradaic/electrostatic reactions on the surface. In the submitted work, the authors have reviewed aerogels having low density nanoporous solids having open fine structures for various energy storage applications including batteries, SCs and fuel cells. The reported review focused on the bio-based aerogel; synthesis, properties and other challenges seen in the applications. This is a significant area of research; however, the work requires lot of clarity. The review in its present form is not suitable and needs some revisions before rendering a final decision.

Thank you for your kind comments.

My specific points are below

Graphene based aerogels are getting wide attention in the area of SCs.

Thank you for your kind comments. Additionally commented in the revised manuscript to enhance clarity. Namely, the reason behind narrower addressing of the graphene based aerogels, is that in majority graphene based aerogels are not bio-based, or are not to-the-point aerogels. Again, we did broaden to put an idea of the investigation of graphene based materials, but we did not focus in graphene based aerogels.

The cost effectiveness of the fabricated supercapacitor can be addressed by cheaper electrode preparation and processing methods using low-cost carbon precursors as source materials.

Thank you for your kind comments. Additionally commented in the revised manuscript to enhance clarity. The focus of this article is on bio-based aerogels, due to their high biodegradability. While low-cost carbon precursors are generally highly relevant, revolving the investigation around them would discriminate the high biodegradability agenda of the paper due to their generally low biodegradability.

The conductivity of the aerogels can be improved by doping with conducting fillers and multiple hetero atoms – which must be included in the appropriate sections.

Thank you for your kind comments. Examples were given in the revised manuscript.

Carbon aerogels derived from these resorcinol formaldehyde-aerogels have a small mesopore surface area, however an especially large micropore area. They provide electrical capacities which are most suitable for their use in supercapacitors.

Thank you for your kind comments. Commented in the revised manuscript to enhance clarity.

What is “solid-state impedance spectroscopy”?

Thank you for you kind comments. Clarified in the revised manuscript.

Materials like lignin, chitosan, alginate, cellulose for electrodes in supercapacitors and battery applications. This could be included and discussed in the synthesis and applications sections of the review article.

Thank you for your kind comments. Additionally commented in the revised manuscript.

Authors have confused between battery (mAh/g) and SC (F/g), in the table it must be clearly identified.

Thank you for your kind comments. Enhanced, see *.

References are not correctly embedded in the text. One simple example, Ramirez et al {no reference}.

Thank you for your kind comments. Corrected.

Figure 6. (a) immediately the reference number [115] – this looks weird.

Thank you for your kind comments. Corrected.

The last application (Fuel Cells) is written very weak.

Thank you for your kind comments. Bio-based aerogels are not used much as the constituents in the fuel cells, so that is something we wanted to emphasise.

Section 6 must include some performance metrics.

Thank you for your kind comments. Examples were given in the revised manuscript.

Section 7 – is it essential? Why materials and methods for review paper? Unless otherwise, authors have included their own original work.

Thank you for your kind comments. We have indeed included our results, but still the section was removed.

Reviewer 2 Report

Comments and Suggestions for Authors

This manuscript is a review article dealing with bio-based aerogels to be used as separators or as a new class of electrically conductive highly porous 3D networks obtained by pyrolysis of porous renewable sources transformed in form of organic aerogels. These bio-based aerogels that are primarily derived from proteins, biopolymers and polysaccharides are proposed for use in energy storage systems. The review topic is quite interesting and original, indeed there exists a notable scarcity of literature papers focusing specifically on the utilization of bio-based aerogels in energy storage device, however some more technical information on pyrolyzed bio-aerogels are needed and should be included to provide readers with a more complete view on this high-potentiality material class. The following few changes are strongly suggested:

- Title is too much detailed, it could be conveniently reduced to a concise form like: “Bio-based aerogels in energy storage systems”.

- A review paper on the topic of organic aerogels from renewable sources should necessarily include all basic information about the described materials. The presented work is not really complete, basic information is lacking and consequently some manuscript improvements are strictly required. Since the bio-aerogel use has been proposed also for electrical scopes (e.g., electrodes for super-capacitors), typical values for basic electrical parameters like the electrical conductivity/resistivity should be provided for each different type of pyrolyzed precursors (cellulose, lignin, chitosan, etc.). Information on device performance and characteristics (e.g., capacitance value) are reported but electrical properties of the used materials must be provided and discussed too. While the process for the conversion of renewable sources to aerogels (sol-gel method) has been enough described, the effect of pyrolysis conditions (time and temperature, type of inert atmosphere, etc.) on these fundamental electrical properties should be given and discussed too. Also morphological information is quite lacking; three SEM micrographs are given in Figure 3 but they are related to the adopted precursors, not to the final carbonized product (e.g., what is the morphology of a lignin-based carbon aerogel?). A detailed description of the generated morphologies could significantly valorize this type of manuscript. Further basic information for porous materials is the surface area development, the apparent density, the porosity, etc. Therefore, the addition of BET with estimated specific surface area is a must for the treated topic. In addition, bio-aerogel electrical conductivity (and other electrical parameters) should be compared with values of standard electrical conductors (metals, graphite, semiconductors, etc.) and comment/discussion about should be also developed. Similarly, a comparison of bio-aerogel specific surface area with values of standard highly porous 3D networks (e.g., lignin-based carbon aerogel compared to resorcinol carbon aerogel) could be discussed too.

- All pictures have a too small size, image/graph details cannot be read or are very difficult to distinguish. Please, try to organize all pictures in a different manner otherwise they are not really useful. For example, a vertical organization of the 11 images in Figure 4 can solve the problem of clearly see their content. The same vertical organization is suggested for the 10 images in Figure 6. The size of the SEM-micrographs in Figure 3 should be increased too. In general, the ratio between text and images/graphs/tables does not seem adequately proportioned, the addition of some more images/graphs/tables is suggested.

- How about the crystalline structure of the carbon-base materials generated by pyrolysis? Are the resulting bio-aerogels graphite-like and therefore highly electrically conductive or similar to some amorphous carbon and therefore with poor conductivity or dielectric behavior? Indeed, some X-ray diffraction (XRD) information should be provided.

- The very short ‘Materials and Methods’ Section placed at the end of the manuscript does not make any sense in a review paper, since it refers only to some of the reviewed preparative procedures and it should be removed.

- More traditional electrically-conductive aerogel materials (i.e., standard materials for electrodes fabrication like metallic aerogels, polyaniline, PVK, PEDOT and other electrically conductive polymers that are competitors of the suggested bio-aerogels) are never cited in the manuscript; the Introduction Section (and/or Section 4) could be advantageously enriched by adding a few up-to-date references about.

- Since ‘bio-aerogels’ represents a very modern topic and papers related to the science/technology of these materials are rapidly appearing in the literature, the manuscript Reference Section should be improved because actually it is not really updated. Authors could imply improve the Reference Section by replacing papers older than 10 years with analogous documents recently appeared.

Author Response

REVIEWER 2

This manuscript is a review article dealing with bio-based aerogels to be used as separators or as a new class of electrically conductive highly porous 3D networks obtained by pyrolysis of porous renewable sources transformed in form of organic aerogels. These bio-based aerogels that are primarily derived from proteins, biopolymers and polysaccharides are proposed for use in energy storage systems. The review topic is quite interesting and original, indeed there exists a notable scarcity of literature papers focusing specifically on the utilization of bio-based aerogels in energy storage device, however some more technical information on pyrolyzed bio-aerogels are needed and should be included to provide readers with a more complete view on this high-potentiality material class.

Thank you for your kind comments.

The following few changes are strongly suggested:

- Title is too much detailed, it could be conveniently reduced to a concise form like: “Bio-based aerogels in energy storage systems”.

Thank you for your kind comments. Corrected in the revised manuscript.

- A review paper on the topic of organic aerogels from renewable sources should necessarily include all basic information about the described materials. The presented work is not really complete, basic information is lacking and consequently some manuscript improvements are strictly required. Since the bio-aerogel use has been proposed also for electrical scopes (e.g., electrodes for super-capacitors), typical values for basic electrical parameters like the electrical conductivity/resistivity should be provided for each different type of pyrolyzed precursors (cellulose, lignin, chitosan, etc.). Information on device performance and characteristics (e.g., capacitance value) are reported but electrical properties of the used materials must be provided and discussed too.

Thank you for your kind comments. Clarified in the revised manuscript.

While the process for the conversion of renewable sources to aerogels (sol-gel method) has been enough described, the effect of pyrolysis conditions (time and temperature, type of inert atmosphere, etc.) on these fundamental electrical properties should be given and discussed too.

Thank you for your kind comments. Clarified in the revised manuscript.

Also morphological information is quite lacking; three SEM micrographs are given in Figure 3 but they are related to the adopted precursors, not to the final carbonized product (e.g., what is the morphology of a lignin-based carbon aerogel?). A detailed description of the generated morphologies could significantly valorize this type of manuscript. Further basic information for porous materials is the surface area development, the apparent density, the porosity, etc. Therefore, the addition of BET with estimated specific surface area is a must for the treated topic. In addition, bio-aerogel electrical conductivity (and other electrical parameters) should be compared with values of standard electrical conductors (metals, graphite, semiconductors, etc.) and comment/discussion about should be also developed. Similarly, a comparison of bio-aerogel specific surface area with values of standard highly porous 3D networks (e.g., lignin-based carbon aerogel compared to resorcinol carbon aerogel) could be discussed too.

Thank you for your kind comments. Clarified in the revised manuscript.

- All pictures have a too small size, image/graph details cannot be read or are very difficult to distinguish. Please, try to organize all pictures in a different manner otherwise they are not really useful. For example, a vertical organization of the 11 images in Figure 4 can solve the problem of clearly see their content. The same vertical organization is suggested for the 10 images in Figure 6. The size of the SEM-micrographs in Figure 3 should be increased too. In general, the ratio between text and images/graphs/tables does not seem adequately proportioned, the addition of some more images/graphs/tables is suggested.

Thank you for your kind comments. Reorganised in the revised manuscript.

- How about the crystalline structure of the carbon-base materials generated by pyrolysis? Are the resulting bio-aerogels graphite-like and therefore highly electrically conductive or similar to some amorphous carbon and therefore with poor conductivity or dielectric behavior? Indeed, some X-ray diffraction (XRD) information should be provided.

Thank you for your kind comments. The crystalline phase evolution and contribution to the electric-dielectric properties has been broadened in the upgraded version of the revised manuscript, though on general basis.

- The very short ‘Materials and Methods’ Section placed at the end of the manuscript does not make any sense in a review paper, since it refers only to some of the reviewed preparative procedures and it should be removed.

Thank you for your kind comments. Removed.

- More traditional electrically-conductive aerogel materials (i.e., standard materials for electrodes fabrication like metallic aerogels, polyaniline, PVK, PEDOT and other electrically conductive polymers that are competitors of the suggested bio-aerogels) are never cited in the manuscript; the Introduction Section (and/or Section 4) could be advantageously enriched by adding a few up-to-date references about.

Thank you for your kind comments. “Competing materials were referenced in the revised manuscript.”

- Since ‘bio-aerogels’ represents a very modern topic and papers related to the science/technology of these materials are rapidly appearing in the literature, the manuscript Reference Section should be improved because actually it is not really updated. Authors could imply improve the Reference Section by replacing papers older than 10 years with analogous documents recently appeared.

Thank you for your kind comments. “Refreshed”; remaining >10 y.o. references are necessary for general context.

Reviewer 3 Report

Comments and Suggestions for Authors

          The paper reviews syntheses, properties and characterization ways of Bio-based aerogels. The bio gels are made from cellulose, lignin, and chitosan source materials. They may be used in numerous practical applications such as energy storage devices such as rechargeable batteries, super-capacitors and fuel cells. One of the recent findings for practical usage is bioaerogel-based anode materials for Lithium-ion batteries in stead of non-renewable carbon materials.

          Aerogels preparation is described in the starting part of the review. The main steps are: Solution-chemistry synthesis shown schematically and Drying processing. Drying schemes such as Ambiental drying, Freeze drying, Supercritical drying are discussed. The pore size and its distribution is known as an important parameter of the  aerogels. The paper reviews the data on several typical pore size and pore size distribution behaviours in bio-based aerogels. Characterization of electrical properties is usually conducted by means of CV–cyclic voltammetry and LSV–linear sweep voltammetry or solid-state and electrochemical impedance spectroscopy. There are presented results on topography and porosity of the main polysaccharide-based bio-aerogels obtained by SEM imaging. A review of possible applications in energy storage Li batteries is presented. The predominant challenges regarding bio-based materials for active electrode use include low electrical conductivity, shrinkage during lithiation and delithiation processes, and uncontrolled lithium dendrite formation. Most of the available scientific articles present results on lignin usage as the material for electrodes used in the Li ion batteries as well as lignin-based electrodes different composite materials. A comparison of summarized performance of bio-aerogels as Electrodes in energy-storage devices is presented. The following device parameters  were considered: specific capacity, coulombic efficiency, number of cycles. The following pure and composite materials were subjected for analysis: lignin, cellulose, resorcinol-formaldehyde, Pd/cellulose, lignin-resorcinol-formaldehyde, TEMPO-cellulose. Applications as energy storage, in supercapacitors is reviewed. Summarized performance of bio-aerogels as separators in energy storage systems. The following device parameters were considered: porosity, pore size, charge/discharge capacity, number of cycles based on the results obtained from lignin, cellulose, cellulose/polyimide and chitosan materials.

               The review would provide a reader with up-to-date information on scientific and practical results within the selected research area. I recommend the review-paper for publication in its present from.

 Regards,  Reviewer

Author Response

REVIEWER 3

The paper reviews syntheses, properties and characterization ways of Bio-based aerogels. The bio gels are made from cellulose, lignin, and chitosan source materials. They may be used in numerous practical applications such as energy storage devices such as rechargeable batteries, super-capacitors and fuel cells. One of the recent findings for practical usage is bioaerogel-based anode materials for Lithium-ion batteries in stead of non-renewable carbon materials.

Aerogels preparation is described in the starting part of the review. The main steps are: Solution-chemistry synthesis shown schematically and Drying processing. Drying schemes such as Ambiental drying, Freeze drying, Supercritical drying are discussed. The pore size and its distribution is known as an important parameter of the  aerogels. The paper reviews the data on several typical pore size and pore size distribution behaviours in bio-based aerogels. Characterization of electrical properties is usually conducted by means of CV–cyclic voltammetry and LSV–linear sweep voltammetry or solid-state and electrochemical impedance spectroscopy. There are presented results on topography and porosity of the main polysaccharide-based bio-aerogels obtained by SEM imaging. A review of possible applications in energy storage Li batteries is presented. The predominant challenges regarding bio-based materials for active electrode use include low electrical conductivity, shrinkage during lithiation and delithiation processes, and uncontrolled lithium dendrite formation. Most of the available scientific articles present results on lignin usage as the material for electrodes used in the Li ion batteries as well as lignin-based electrodes different composite materials. A comparison of summarized performance of bio-aerogels as Electrodes in energy-storage devices is presented. The following device parameters  were considered: specific capacity, coulombic efficiency, number of cycles. The following pure and composite materials were subjected for analysis: lignin, cellulose, resorcinol-formaldehyde, Pd/cellulose, lignin-resorcinol-formaldehyde, TEMPO-cellulose. Applications as energy storage, in supercapacitors is reviewed. Summarized performance of bio-aerogels as separators in energy storage systems. The following device parameters were considered: porosity, pore size, charge/discharge capacity, number of cycles based on the results obtained from lignin, cellulose, cellulose/polyimide and chitosan materials. The review would provide a reader with up-to-date information on scientific and practical results within the selected research area. I recommend the review-paper for publication in its present from. Regards, Reviewer.

Thank you for your kind comments.

Reviewer 4 Report

Comments and Suggestions for Authors

The topic of the submitted review article is interesting. However, the article suffers from unclear writing and poor illustrations. The text should work with the illustrations but this does not seems to be the case here. In addition, many of the Figures are too small. The authors should make some schematic illustrations that provide generalizations and describe important concepts. Such figures would be of higher value than just copying plots from other articles (as most of the Figures are in the submitted version). 

 It is written:

“Therefore, bio-based aerogels display several typical pore size and pore size distribution behaviours:

• if used for bone tissue engineering, the pore size of the used sample should be be tween 100 and 350 um allowing bone ingrowth and regeneration , while distribution should be bimodal; containing as well the mesopores in size between 20 and 50 nm, ensuring good mechanical properties of the sample

• when commenting on drug delivery applications, pore size is depending highly on the drug which should be delivered, but concerning distribution, it should be uniform,”

I suggest the authors remove the discussion related to bone tissue engineering and drug delivery. The article gets more coherent if you focus on the key topic.

Maybe the authors can provide an improved and more focused introduction by focusing on aerogels for energy storage? 

Comments on the Quality of English Language

The beginning of the article should be improved. I provide some detailed comments here, so that the authors can get a clear idea of possible improvements.

It is written: “Aerogels can be categorized into two main types, inorganic and organic, with further subdivision based on the materials used in their gel structure design [1]. Interestingly, recent attention seems to be going in the direction of biodegradable and bio-based polymers due to their potential to reduce environmental impact [2]. Namely, these porous materials offer versatile properties and can be tailored through specific synthesis routes for a broad range of applications. For instance, polysaccharide-based aerogels have been utilized for environmental engineering, construction, medical practice, packaging, electrochemistry, and other [3]. However, bio-based aerogels are primarily employed as adsorbent materials and drug-delivery systems [4].”

“Aerogels can be categorized into two main types, inorganic and organic, with further subdivision based on the materials used in their gel structure design [1]”

OK, but as the title includes “Bio-based aerogels” it is pretty obvious that there will not be much discussion of inorganic aerogels. (or are you having carbonzed structures in mind? )

“Interestingly, recent attention seems to be going in the direction of biodegradable and bio-based polymers due to their potential to reduce environmental impact [2]”

OK

“Namely, these porous materials offer versatile properties and can be tailored through specific synthesis routes for a broad range of applications.”

As written this statement seems to concern the biodegradable and bio-based polymers mentioned in the preceding sentence. However, this does not makes sense as these polymers often occur in other forms than aerogels. The statement probably refers to the first sentence, but if so – why insert the sentence about bio-based polymer in between? This only creates confusion. There are many ways to fix the problem, but the provided version is unnecessarily confusing.

“For instance, polysaccharide-based aerogels have been utilized for environmental engineering, construction, medical practice, packaging, electrochemistry, and other”

OK

“However, bio-based aerogels are primarily employed as adsorbent materials and drug-delivery systems [4]”

This is confusing, polysaccharides are certainly bio-based. The statement does not match well with the preceding sentence.

“Their advantages include mesoporosity and high specific surface area, as well as their biocompatibility and biodegradability, which are particularly advantageous in the nowadays-relevant context of environmental friendliness.”

The first part (mesoporosity and high specific surface area) is true for many types of aerogels. However, the way it is written one gets the impression that these are unique features of bio-based aerogels.

Author Response

REVIEWER 4

The topic of the submitted review article is interesting. However, the article suffers from unclear writing and poor illustrations. The text should work with the illustrations but this does not seems to be the case here. In addition, many of the Figures are too small. The authors should make some schematic illustrations that provide generalizations and describe important concepts. Such figures would be of higher value than just copying plots from other articles (as most of the Figures are in the submitted version).

Thank you for your kind comments. Reorganised and resized accordingly in the revised manuscript.

It is written:

“Therefore, bio-based aerogels display several typical pore size and pore size distribution behaviours:

  • if used for bone tissue engineering, the pore size of the used sample should be between 100 and 350 um allowing bone ingrowth and regeneration , while distribution should be bimodal; containing as well the mesopores in size between 20 and 50 nm, ensuring good mechanical properties of the sample
  • when commenting on drug delivery applications, pore size is depending highly on the drug which should be delivered, but concerning distribution, it should be uniform,”

I suggest the authors remove the discussion related to bone tissue engineering and drug delivery. The article gets more coherent if you focus on the key topic.

Thank you for your kind comments. Upgraded accordingly in the revised manuscript.

Maybe the authors can provide an improved and more focused introduction by focusing on aerogels for energy storage?

Thank you for your kind comments. Upgraded accordingly in the revised manuscript.

Comments on the Quality of English Language: The beginning of the article should be improved. I provide some detailed comments here, so that the authors can get a clear idea of possible improvements. It is written: “Aerogels can be categorized into two main types, inorganic and organic, with further subdivision based on the materials used in their gel structure design [1]. Interestingly, recent attention seems to be going in the direction of biodegradable and bio-based polymers due to their potential to reduce environmental impact [2]. Namely, these porous materials offer versatile properties and can be tailored through specific synthesis routes for a broad range of applications. For instance, polysaccharide-based aerogels have been utilized for environmental engineering, construction, medical practice, packaging, electrochemistry, and other [3]. However, bio-based aerogels are primarily employed as adsorbent materials and drug-delivery systems [4].”

“Aerogels can be categorized into two main types, inorganic and organic, with further subdivision based on the materials used in their gel structure design [1]”

OK, but as the title includes “Bio-based aerogels” it is pretty obvious that there will not be much discussion of inorganic aerogels. (or are you having carbonzed structures in mind? )

“Interestingly, recent attention seems to be going in the direction of biodegradable and bio-based polymers due to their potential to reduce environmental impact [2]”

OK

“Namely, these porous materials offer versatile properties and can be tailored through specific synthesis routes for a broad range of applications.”

As written this statement seems to concern the biodegradable and bio-based polymers mentioned in the preceding sentence. However, this does not makes sense as these polymers often occur in other forms than aerogels. The statement probably refers to the first sentence, but if so – why insert the sentence about bio-based polymer in between? This only creates confusion. There are many ways to fix the problem, but the provided version is unnecessarily confusing.

“For instance, polysaccharide-based aerogels have been utilized for environmental engineering, construction, medical practice, packaging, electrochemistry, and other”

OK

“However, bio-based aerogels are primarily employed as adsorbent materials and drug-delivery systems [4]”

Thank you for your kind comments. Upgraded accordingly in the revised manuscript.

This is confusing, polysaccharides are certainly bio-based. The statement does not match well with the preceding sentence.

“Their advantages include mesoporosity and high specific surface area, as well as their biocompatibility and biodegradability, which are particularly advantageous in the nowadays-relevant context of environmental friendliness.”

The first part (mesoporosity and high specific surface area) is true for many types of aerogels. However, the way it is written one gets the impression that these are unique features of bio-based aerogels.

Thank you for your kind comments. Upgraded accordingly in the revised manuscript.

Round 2

Reviewer 4 Report

Comments and Suggestions for Authors

The authors have made a thorough revision.

I just have a small comment regarding Figure 2:

there is a (c) in Figure 2a and no scale bar in Figure 2c